# Towards Efficient Multi-Agent Learning Systems

Kailash Gogineni, Peng Wei, Tian Lan and Guru Venkataramani

The George Washington University, Washington, DC, USA

E-mail: {kailashg26, pwei, tlan, guruv}@gwu.edu

*Abstract*—**Multi-Agent Reinforcement Learning (MARL) is an increasingly popular domain for modeling and controlling multiple large-scale autonomous systems. Existing multi-agent learning implementations typically involve intensive computations in terms of training time and power requirements arising from large observation-action space and a huge number of training steps. Therefore, a key challenge is understanding and characterizing the computationally intensive functions in several popular classes of MARL algorithms during their training phases. Our preliminary experiments reveal new insights into the key modules of MARL algorithms that limit their adoption in real-world systems. We explore neighbor sampling strategy to improve the cache locality and observe performance improvement ranging from $26.66\%$ (3 agents) to $27.39\%$ (12 agents) for the computationally intensive mini-batch sampling phase. Additionally, we demonstrate that improving the cache locality leads to an end-to-end training time reduction of $10.2\%$ (for 12 agents) compared to existing multi-agent algorithms without significant degradation in the mean reward.**

*Index Terms*—**Multi-Agent Systems, Performance Analysis, Reinforcement Learning, Performance Optimization**

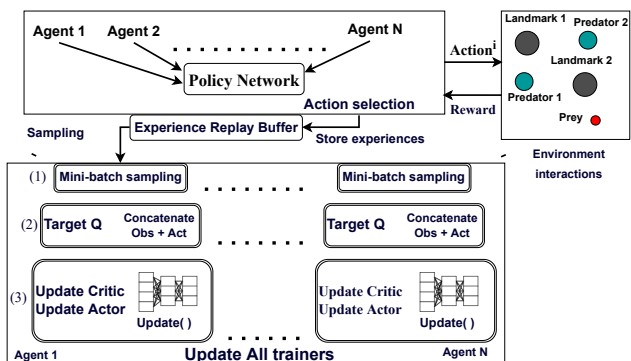

Fig. 1: Overview of our multi-agent decentralized actor, centralized critic approach (Competitive environment).

## I. INTRODUCTION

Reinforcement Learning (RL) has recently made great progress in many applications, including Atari games [1], aviation systems [2], and robotics [3]. Specifically, RL frameworks fit in the context of addressing problems that involve sequential-decision making where the agent needs to take actions in an environment to maximize the cumulative rewards. In RL, the quality of state-action pairs is evaluated using a reward function, and the transition to a new state depends on the current state and action [4]. The function that determines the action from the state is known as a policy. The function representing the reward estimates is known as the value function.

Multi-agent systems [4] have shown excellent performance among various multi-player games [5] where there is significant sharing of observations between the agents during training, and the joint actions among these agents could affect the environment dynamically. In MARL, several agents simultaneously explore a common environment and perform competitive (e.g., Predator-prey) and cooperative (e.g., Cooperative navigation) tasks [6]. All the observations are shared in the cooperative setting, and the training is performed centrally. In contrast, each agent aims to outperform its enemies in a competitive setting. As a result, MARL training involves several *computationally-challenging* and *memory-intensive* tasks that deal with dynamically changing environments.

In this paper, we performed a workload characterization study to understand the *performance-limiting functions* on well-known model-free MARL frameworks [6], [7] implemented using actor-critic methods with state spaces that are usually very large. We analyze different MARL training phases where the actor and critic networks are responsible for policy and value functions. The critic tries to learn a value function given the policy from the actor, while the actor can estimate the policy gradient based on the approximate value function that the critic provides.

As shown in Figure 1, each agent in the environment has its own actor network, which outputs the action of an agent given its observation (*Action selection*). During the *mini-batch sampling* phase, each agent $i$ collects the historical transition data of all other agents stored within the *Experience Replay Buffer*. The sampling approach enables the algorithm to reuse the transition data for updating the current policy. Each agent has a centralized critic, which outputs the Q-value using the joint observation-action space of all other agents. During *Update all trainers* phase, both the actor and critic networks are updated after the *target Q calculation* and *sampling phase*.

The main contributions of our paper are the following:

- We systematically perform a hardware-software performance analysis within the training phases of Multi-agent systems. We present key insights into the performance bottlenecks confronting several key MARL algorithms from a systems perspective.
- We explore a neighbor sampling strategy to improve the locality of data access within the *mini-batch sampling* phase. Our preliminary experiments provide performance improvement ranging from $26.66\%$ (3 agents) to $27.39\%$ (12 agents) in the sampling phase training run-time. Additionally, we achieve $10.2\%$ (12 agents) end-to-end

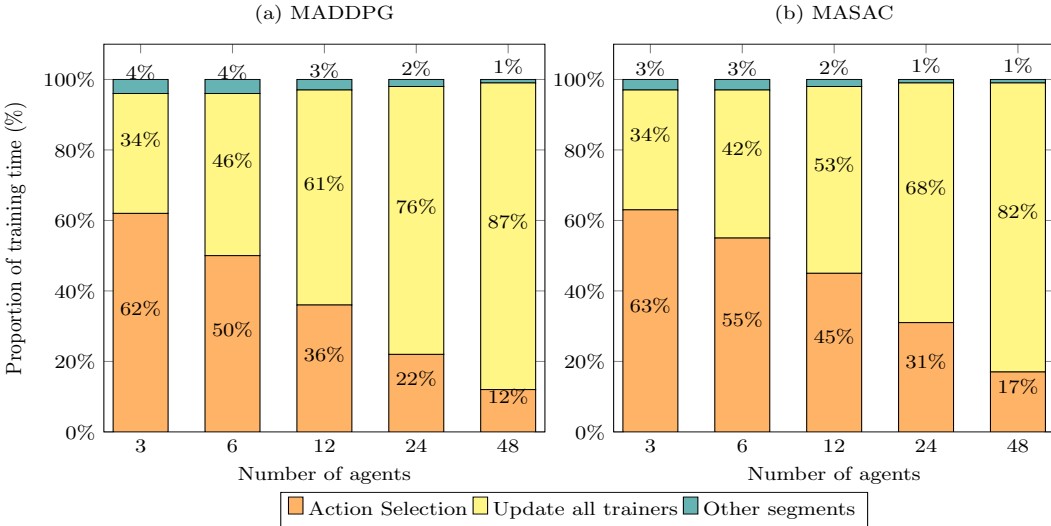

Fig. 2: Training time breakdown on Ampere Architecture RTX 3090 for the MARL workloads (MADDPG [6] & MASAC [7]) in multi-agent settings. The simulated multi-agent particle environment is Predator-Prey.

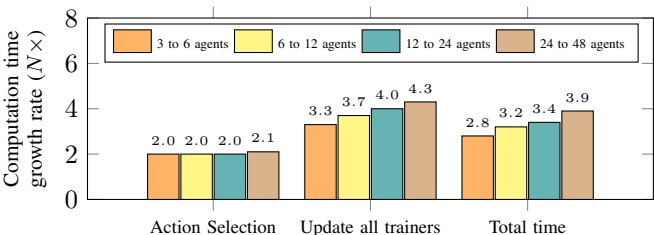

Fig. 3: Computation time growth in MARL modules averaged across the two MARL frameworks (MADDPG [6] & MASAC [7]).

training time reduction compared to the state-of-the-art multi-agent algorithms.

## II. MOTIVATION

In multi-agent systems, the training phase is performance and memory-intensive as the agents must collaborate and coordinate to maximize a shared return [8]. Many real-world applications, such as robot fleet coordination [9] and traffic light control [10], are modeled as multi-agent problems, but they become intractable with the growing number of agents due to the intensive computations required to estimate other agents' policies at each state and a huge amount of neural network parameters. This limits their adoption in real-world systems and limits applications only to scenarios with a few agents [11], [12]. Figure 2 shows the run-time breakdown of the training phase[1]. *Update all trainers* contributes to $\approx 35\%$ to $\approx 85\%$ of the training time as the number of MARL agents grows from 3 to 48. This is mainly due to two reasons: ① In MARL, each agent has its own actor and critic networks.

[1]We omit the *agents interactions* phase since it primarily depends on environment complexity.

Each agent must randomly collect a batch of transitions from all other agents to update the critic and actor networks. ② The dynamic memory requirements of observation and action spaces also grow quadratically due to each agent having to coordinate with other agents towards sharing their observations and actions. *Action selection* phase scales linearly with the number of agents (Figure 3). This is because, in *Action selection*, agents consider individual policies to obtain local actions. *Other segments* include experience collection, reward collection, and policy initialization, and they add a negligible overhead.

## III. BACKGROUND

Typically, MARL settings with $N$ agents is defined by a set of states, $S = S_1 \times ... \times S_N$, a set of actions $A = A_1 \times ... \times A_N$. Each agent selects its action by using a policy $\pi_{\theta_i} : O_i \times A_i \to [0,1]$. The state transition ($T : S \times A_1 \times A_2 \times ... \times A_N$) function produces the next state $S'$, given the current state and actions for each agent. The reward, $R_i : S \times A_i \to \mathbb{R}$ for each agent is a function of global state and action of *all other agents*, with the aim of maximizing its own expected return $R_i = \sum_{t=0}^{T} \gamma^t r_i^t$, where $\gamma$ denotes the discount factor and $T$ is the time horizon. For this, we use the actor-critic methods such as MADDPG [6], MASAC [7].

MADDPG [6] is centralized training and decentralized execution (CTDE) algorithm mainly designed for mixed environments. Each agent learns an individual policy that maps the observation to its action to maximize the expected return, which is approximated by the critic. MADDPG lets the critic of agent $i$ to be trained by minimizing the loss with the *target Q-value* and $y_i$ using $\mathcal{L}(\theta_i) = \mathbb{E}_D[(Q_i(S, A_1, ...A_n) - y_i^2]$, and $y_i = r_i + \gamma \overline{Q}_i(S', A'_1, ...A'_n)_{a'_j = \overline{\pi}(o'_j)}$, where $S$ and $A_1, ...A_n$ represent the joint observations and actions respectively. $D$ is the experience replay buffer that stores the *observations, actions, rewards, and new observations* samples of all agents

obtained after the training episodes. The critic networks are augmented with states and actions of all agents to reduce the variance of policy gradients and improve performance. The MARL framework has four networks- actor, critic, target actor, and target critic. $\overline{Q}_i$ and $\overline{\pi}(o'_j)$ are the target networks for the stable learning of critic ($Q_i$) and actor networks. The target actor estimates the next action from the policy using the state output by the actor network. The target critic aggregates the output from the target actor to compute the target Q-values, which helps to update the critic network and assess the quality of actions taken by agents. The target networks are created to achieve training stability. Note that the updating sequence of networks in the back-propagation phase is critics, actors, then the target networks.

Similar to MADDPG, the centralized critic is introduced in Soft Actor-Critic (SAC [7]) algorithm. MASAC uses the maximum entropy RL, in which the agents are encouraged to maximize the exploration within the policy. MASAC assigns equal probability to nearly-optimal actions which have similar state-action values and avoids repeatedly selecting the same action. This learning trick will increase the stability, policy exploration, and sample efficiency [7], [13].

## IV. EVALUATION SETUP

**Benchmark.** Table I provides the behavior of selected Multi-agent Particle Environments (MPE [6]). We profile and characterize two state-of-the-art MARL algorithms, MADDPG and MASAC. A two-layer ReLU MLP parameterizes the actor and critic networks with 64 units per layer, and the mini-batch size is 1024 for sampling the transitions. In our experiments, we use Adam optimizer [14] with a learning rate of 0.01, maximum episode length as 25 (max episodes to reach the terminal state), and $\tau = 0.01$ for updating the target networks. $\gamma$ is the discount factor which is set to 0.95. The size of the replay buffer is 1 million, and the entropy coefficient for MASAC is 0.05. The network parameters are updated after every 100 samples are added to the replay buffer.

TABLE I: Multi-agent particle environment.

| Environment | Details |
|---|---|
| Cooperative navigation | $N$ agents move in a cooperated manner to reach $L$ landmarks and the rewards encourages the agents get closer to the landmarks. |
| Predator-Prey | $N$ predators work cooperatively to block the way of $M$ fast paced prey agents. The prey agents are environment controlled and they try to avoid the collision with predators. |

**Profiling Platform.** MARL algorithms are implemented with state-of-the-art CPU-GPU compatible TensorFlow-GPU (v2.11.0). The server runs on Ubuntu Linux 20.04.5 LTS operating system with CUDA 9.0, cuDNN 7.6.5, PCIe Express® v4.0 with NCCL v2.8.4 communication library. The machine supports Python 3.7.15, TensorFlow-Slim (v1.1.0) and OpenAI GYM (v0.10.5). All the workloads are profiled on single Nvidia GeForce RTX 3090 Ampere Architecture with Perf [15] and NVProf to profile hardware performance counters for performance analysis. Finally, we trained for 60K

episodes using default hyper-parameters recommended by the algorithms.

## V. EVALUATION

In this section, we first present an overview of our MARL profiling results. Then, we study the computationally dominant functions within *Update all trainers*: *Mini-batch sampling, Target Q calculation*, and *Q loss & P loss* and present our results in the competitive setting (predator-prey) to understand the key factors limiting MARL in large-scale systems.

Figure 4 shows the breakdown between the modules, *Mini-batch sampling, Target Q calculation, Q loss, and P loss* that contribute 63%, 24%, 6.5%, and 6% to the overall computation time averaging across different workloads for 48 agents.

### A. Mini-batch sampling

Our experimental results in Figure 4 show that mini-batch sampling is the largest time-consuming phase within the *Update All Trainers* module. The behavior is also consistent with scaling in other critical hardware performance metrics: *dTLB load misses*-$3.9\times$ (growth rate from $3-6$ agents) and *cache misses*-$3.9\times$ (growth rate from $3-6$ agents).

*Mini-batch sampling* phase is dominated by the collection of random samples from all other agents' replay buffers and updates the parameters of its actor and critic networks. Note that the agent replay buffers are kept separate from each other to capture their past transitions. For each time-step, agent $i$ draws a random index set $\{L_1, L_2, ...., L_K\}$ ($K$ is the mini-batch size), and first selects $L_1$ to perform a memory lookup in the experience replay buffer to retrieve the corresponding transition and store it in the individual agent buffer. This operation grows as a function of the number of agents, $N$, since it is repeated on all $N$ agents. The sampling stage exhibits random memory access patterns and cannot exploit the cache reuse due to randomness in the indices for each agent between the iterations. In cooperative navigation (*simple spread* [6]), we observe similar bottlenecks since all the agents are trained together to reach the landmarks while avoiding collisions with each other.

### B. Target Q calculation

The *Target Q calculation* phase is the second largest time-consuming phase within *Update All Trainers* (Figure 4). In this function, each agent performs the *next action calculation, target Q next, and target Q values* as a function of all other agents' joint observation-action space. To calculate the *next action*, the agent $i$ uses its policy network to determine *next action-a'* from the *next state-S'*. In this phase, each agent's policy network involves multiplications with input-weight matrix and additions resulting in performance impact. The obtained a' and S' data are aggregated and concatenated into a single vector in order to compute the *target Q next* amongst the cooperating agents. The input space (dimension) for the *Q-function* increases quadratically with the number of agents [16]. The target critic values for each agent $i$ are computed using *target Q next* values from the target actor

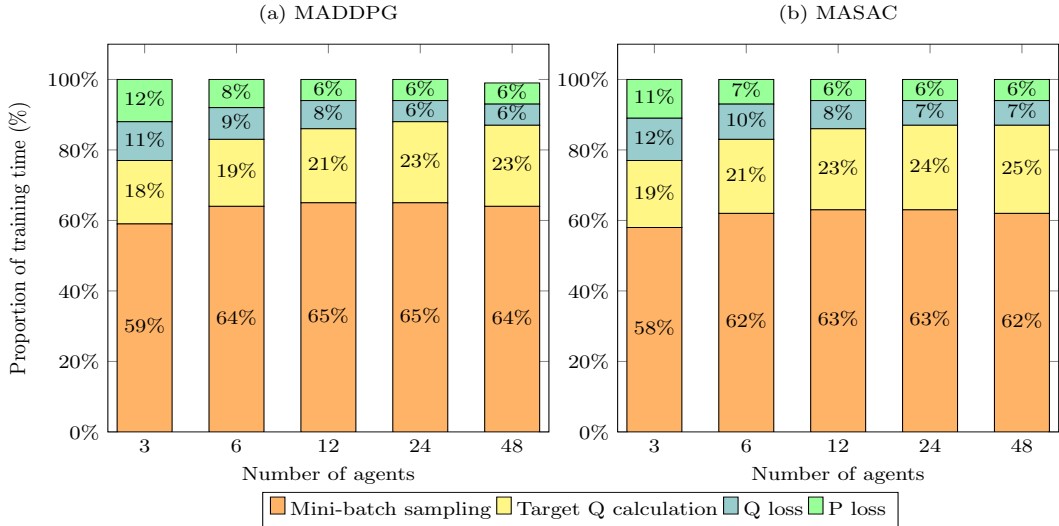

Fig. 4: Training time breakdown on Nvidia Ampere Architecture RTX 3090 within *Update all trainers* on two different MARL workloads (MADDPG & MASAC) in multi-agent settings under the Predator-Prey environment.

network. We note that each agent has to read other agents' policy values; as such, for $N$ agents, there is $N \times (N-1)$ memory lookup operations corresponding to the *next action-a'*.

### C. Back-propagation - Q loss & P loss

Back propagation stage is dominated by the execution of two networks: ① *critic network* computes the mean-squared error loss between the target critic and critic networks, and ② the *actor network* is updated by minimizing the Q values (computed by the critic network). This is because as the number of agents increases, the trainable parameters increase, and $N$ policy and $N$ critic networks are built for all $N$ agents, which incurs extra time to update the weights for each agent. For each update, we sample the random mini-batch of transitions (1024 in our studies) from the replay buffer of each agent $i$ across all agents and then perform gradient descent on the critic and actor networks.

### VI. NEIGHBOR SAMPLING STRATEGY

From our analysis so far, it can be concluded that the *mini-batch sampling* phase dominates *Update all trainers* when the number of agents scales linearly. Moreover, fetching the transition data from the far away memory locations significantly affects the overall training time. Among all the hardware metrics, cache misses suffer from the worst scaling factor (at least $3.9\times$ for 3-6 agents). Therefore, with the support of loop-level optimization, we explore optimizations that can improve the locality and overall MARL performance. To address this issue, we propose a loop-level optimization approach while accessing the transition data in the *mini-batch sampling* phase.

The idea of this approach is to eliminate the computation issues arising due to fetching the data from far away memory locations based on random indices. We investigate the

---

**Algorithm 1** Neighbor Sampling Strategy

    **Input:** Mini-batch indices $MB\_idx$; replay buffer $\mathcal{D}$ with size $d$; micro-batch size $n$

    **Output:** Mini-batch transitions

1: Initialize $obs\_t$, $actions\_t$, $rewards\_t$, $obs\_next\_t$, $terminal\_state\_t$, at time $t \leftarrow \{\emptyset\}$
2: **for** $i$ in $MB\_idx$ **do**
3:     $\alpha \leftarrow [j | max(0, i-n) \leq j < min(d, i+n+1), j \neq i]$
    ▷ $\alpha$ includes all indices in the range $(i-n)$ to $(i+n)$, excluding the current index, and also ensuring not to go below 0 or exceed the length of Replay buffer $\mathcal{D}$
4:     **for** $k$ in $\alpha$ **do**
5:         **if** $k \in \mathcal{D}$ **then**
6:             $obs, act, next\_obs, rew, done \leftarrow unpack(\mathcal{D}[k])$
    ▷ Append these unpacked transition data to the corresponding lists $obs\_t$, $actions\_t$, $rewards\_t$, $obs\_next\_t$, $terminal\_state\_t$
7:         **end if**
8:     **end for**
9:     **if** $len(obs\_t) \geq$ size of $(MB\_idx)$ **then**
10:         $break$
11: **return** $obs\_t$, $actions\_t$, $rewards\_t$, $obs\_next\_t$, $terminal\_state\_t$
    ▷ Return the corresponding lists converted into NumPy arrays
12:     **end if**
13: **end for**

---

neighbor sampling optimization in MADDPG, where we collectively capture the neighbor transitions of an index $i$ to enable faster data access on a given hardware. Intuitively, at each index $i$, we group the neighbor indices into a single micro-batch and extract the data in a locality-aware memory access order to efficiently sample the transitions.

**Neighbor Sampling Strategy.** Algorithm 1 shows how the *mini-batch sampling* phase selects the neighboring transitions for a random index $i$. We initialize replay buffer $\mathcal{D}$, micro-batch size $n$. The algorithm iterates over the mini-batch indices to collect transition data for every index. We modify the loop to accumulate a micro-batch of transitions spanning a range of $n$ neighbors surrounding the current index $i$, i.e., for every index $i$, we check if $i$ is within the limits of replay buffer $\mathcal{D}$. If so, we capture the buffer indices from $i - n$ to $i + n$ based on the micro-batch size $n$ and return a list of neighbors $\alpha$ (line 3). We perform an array access for all the indices in $\alpha$, and the output vectors are unpacked and stored as individual vectors in the experience replay tuple consisting of *obs, act, next_obs, rew, done* (line 5). These individual vectors are appended to their corresponding parents lists *obs_t, actions_t, rewards_t, obs_next_t, terminal_state_t* (line 6). Finally, all the parent lists which contain the transition data at time-step $t$ are converted as vectors. The loop terminates when the mini-batch size is reached (equal to the size of $MB\_idx$) (line 9).

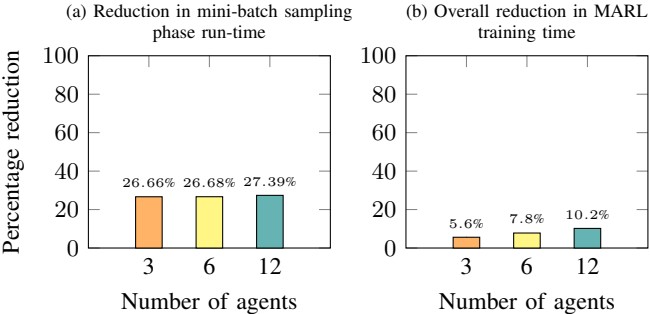

Fig. 5: (a) Percentage reduction in training time for the *mini-batch sampling* phase for 3, 6 and 12 agents (MADDPG). (b) Percentage reduction in the *total training time* when the number of agents are scaled by $2\times$ for MADDPG. The environment test-bed is Predator-Prey and the *micro-batch size=3*

Overall, our neighbor sampling optimization improves performance through leveraging the spatial locality and achieves training time reduction ranging from $26.66\%$ (3 agents) to $27.39\%$ (12 agents) during the computationally intensive mini-batch sampling (Figure 5). In addition, we achieve an end-to-end training time reduction of $10.2\%$ for 12 agents. While studying this optimization; we ensure no significant degradation in the mean episode reward.

## VII. Discussion and Related Work

Hardware-Software acceleration techniques in RL have been studied in recent years [17]–[20]. For example, to accelerate RL training from the software standpoint, prior works have shown that half-precision (FP16) quantization can yield significant performance benefits and improve the hardware efficiency while achieving adequate convergence [21]. Other relevant approaches include QuaRL [22], where quantization is applied

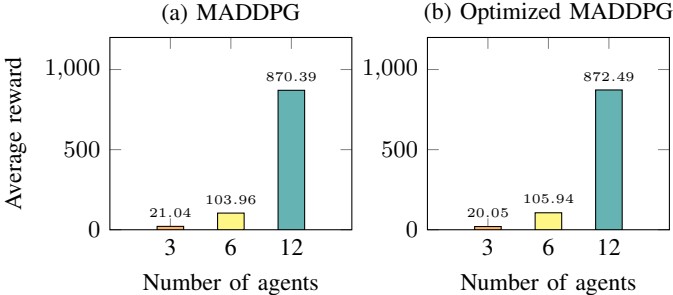

Fig. 6: (a) Average of mean episode rewards of all the agents trained for 60,000 episodes for MADDPG. (b) Average of mean episode rewards all the agents trained for 60,000 episodes after the neighbor sampling optimization for MADDPG. The environment test-bed is Predator-Prey and the *micro-batch size=3*.

to speed up the RL training and inference phases. QuaRL experimentally demonstrated that quantizing the policies to $\leq 8$ bits led to substantial speedups in the training time compared to full precision training. All of the prior works differ from our work as they apply quantization to single-agent RL algorithms or neural networks. Further, we explore the neighbor sampling optimization to improve the efficiency of *mini-batch sampling* phase.

Prior studies, like FA3C, have focused on hardware acceleration in multiple parallel worker scenarios, where each agent is controlled independently within their environments using single-agent RL algorithms [18]. In contrast, we seek to systematically understand the performance-limiting functions in multi-agent systems, where the agents collaborate in a single shared environment. Agents in such MARL settings usually have high visibility of one another (leading to large space and action spaces). Apart from the methods that focus on accelerating multiple parallel worker scenarios, other approaches use a *transition data-reuse optimization* to improve the cache locality and training time [23]. The authors experimentally demonstrated that applying the optimal prioritization scheme proposed by MAC-PO [24] on multi-agent learning problems and repeatedly reusing the transition data with higher weights improves the training efficiency.

In MARL settings where each agent needs to interact with its neighbor agents, especially in complex environments with lots of observations and huge action spaces, computational bottlenecks may be alleviated using architectural primitives implementing selective attention [13], [25], [26]. As the number of agents increases, the hardware techniques such as near-memory computing could help to perform *mini-batch sampling* efficiently. For the input to critic networks, multi-level data compression [27]–[29] techniques on a targeted group of agents may be used based on their importance in the environment. Also, the cache misses during *mini-batch sampling* phase indicate competition for the LLC cache, which may be addressed through smart cache allocation strategies.

Other modules, such as *next action calculation, environment interactions, and action selection phases*, may also benefit from the custom acceleration of key modules.

## VIII. CONCLUSION AND FUTURE WORK

In this work, we present an end-to-end characterization of several popular Multi-Agent Reinforcement Learning algorithms and, in particular, explore the locality-aware neighbor indexing optimization. We find that the *Update all trainers* dominates the training process of MARL algorithms and scales super linearly with the number of agents. Our experimental analysis presents key insights into the modules that are the driving factors behind computational bottlenecks. We also propose a loop-level optimization approach for accessing transition data in the *mini-batch sampling* phase. The proposal achieves performance improvement from $26.66\%$ (3 agents) to $27.39\%$ (12 agents) within the *mini-batch sampling* phase. In future work, we will investigate various efficient sampling strategies and design a hardware-friendly architecture to efficiently fetch the transitions in large-scale MARL.

## ACKNOWLEDGMENT

This research is based on work supported by the National Science Foundation under grant CCF-2114415. We would also like to thank the reviewers for their valuable feedback.

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
