# OpenReview forum: "Towards Efficient Multi-Agent Learning Systems"
_iscaconf.org/ISCA/2023/Workshop/ASSYST — ASSYST Oral_

### Official Review · Reviewer_SGJA · 2023-05-04
**Profiling result is interesting, but the optimization is less so**

**Rating:** 5
**Confidence:** 2

**Review:**

Multi-Agent Reinforcement Learning (MARL) is an interesting emerging workload to optimize for. The nature of quadratically increased computation and memory requirement in this multi-agent scenario is a challenging and requires more study.

This paper provides an detailed profiling and workload analysis for two popular MARL framework, MADDPG and MASAC, and show how the execution time breakdown changes when the number of agent in the system changes. It is interesting to see the only the update trainer phase has a quadratic increase in computation, while the action selection does not. It also makes perfect sense that given only the update trainer phase increase quadratically, once the number of agents increases, the training phase is dominated by it.

The paper also mentioned that in the two different scenarios, the prey-predator case and the cooperative case, the compute behavior is similar. This observation is also useful and insightful, since the MARL system can be use to model various environment, and an effective HW/SW solution can be effective in all cases.

While the paper provides interesting insight on the profiling results, the optimization is less insightful. The author claims that the issue is the random accesses to various agent states, and proposes a solution that generates regulated memory accesses by limiting the range of neighbor sampling. Even though the paper provides some proofs to show that this optimization does not alter the effectiveness of the system, it is not convincing as one can always construct a edge case where looking at the closest N neighbor will give suboptimal results.

Moreover, it is not so clear the connection between this paper and transformer models, which is the focus of the workshop. The audience and attendees might benefit more if the paper can show how transformer models can be applied to MARL systems, and if it changes the profiling result. For example, the balance between action selection and update trainer could be different, if the action selection network is a transformer network that might also have a quadratic scaling in the attention layer (e.g., the number of agents is used as the sequence length).

Overall this is an interesting paper, but there is potential room for improvement.

**Review (Strengths/Weaknesses):**

Strength:

1. Important and interesting workload

2. Detailed analysis and profiling result

Weaknesses

1. Optimization for cache locality is not very convincing

2. Less connection to the transformer models, which is the main focus on the workshop.

**Reviewer Expertise:**

Knowledgeable: I used to work in this area and/or I try to keep up with the literature but might not know the latest developments.

---

### Official Review · Reviewer_iwFU · 2023-05-08
**Good work with rooms for improvements**

**Rating:** 6
**Confidence:** 3

**Review:**

The paper analyzes the computationally intensive functions in several popular classes of Multi-Agent Reinforcement Learning (MARL) algorithms. In addition, the authors present a neighbor sampling strategy to improve cache locality and reduce the training time of MARL algorithms while maintaining the mean reward.

The paper begins with a motivation for studying MARL algorithms, which can model and control multiple large-scale autonomous systems. However, the authors note that the expensive computations required during the training phase limit their adoption to real-world systems. They present a performance analysis of MARL algorithms and identify Update all trainers as the most time-consuming phase. The paper then provides a detailed description of the MARL algorithm and its various training phases.

The authors present their main contribution, a neighbor sampling strategy to improve the locality of data access within the mini-batch sampling phase. They provide preliminary experiments that show a performance improvement ranging from 26.66% to 27.39% during the sampling phase training run-time and a 10.2% end-to-end training time reduction compared to the state-of-the-art multi-agent algorithms.

The experimental results indicate that mini-batch sampling is the most time-consuming phase within the Update All Trainers module, with scaling in other critical hardware performance metrics such as dTLB load misses, and cache misses. The target Q calculation phase is the second largest time-consuming phase, while the back-propagation stage is dominated by the execution of two networks: a critic network that computes the Mean-Squared Error loss and an actor network that is updated by minimizing the Q values computed by the critic network.

To optimize the mini-batch sampling phase, the paper proposes a neighbor sampling strategy that groups the neighbor transitions of an index to enable faster data access on a given hardware. Algorithm 1 shows how the mini-batch sampling phase selects the neighboring transitions for a random index i. The first loop maintains the indices' random accesses. The second loop extracts the neighbor transitions from the experience replay buffer and stores them as individual vectors in the experience replay tuple. Finally, all the transitions are individually accumulated as numpy vectors, and the batch size b is returned when the size of observations becomes full.

The neighbor sampling strategy improves the locality and overall multi-agent reinforcement learning performance by eliminating the computation issues arising due to fetching data from far away memory locations based on random indices. The proposed approach is tested on MAPPDG, and the experimental results show that it significantly reduces the mini-batch sampling time and cache misses, leading to faster convergence and better performance.

Overall, the paper provides a valuable contribution to the field of MARL by identifying performance bottlenecks in several key MARL algorithms and presenting a solution to improve their training time. The paper is well-written, and the analysis is presented in a clear and concise manner. However, the preliminary experiments presented could be improved with a more extensive evaluation to ensure the generalizability of the results.

**Review (Strengths/Weaknesses):**

While the proposed technique is useful, the paper has some limitations and shortcomings that could be addressed in future work:

- The experiments conducted in the paper are limited to a few benchmark problems, and it is unclear how the proposed technique would perform on more complex real-world problems.
- The paper does not provide a detailed analysis of the computational costs associated with different parts of the proposed method, which makes it difficult to assess the overall efficiency of the approach.
- The paper does not provide a thorough comparison with other related work, such as techniques that use attention mechanisms or multi-level data compression.

To complement the work, further analyses could be conducted to address these limitations. For example, a more detailed analysis of the computational costs associated with different parts of the proposed method could help identify areas for further optimization. Additionally, a comparison with other related work could provide a better understanding of the strengths and weaknesses of the proposed approach. Finally, experiments on more complex real-world problems could demonstrate the applicability of the proposed method in practical settings.

**Reviewer Expertise:**

Knowledgeable: I used to work in this area and/or I try to keep up with the literature but might not know the latest developments.

---

### Official Review · Reviewer_1BU9 · 2023-05-12
**Towards Efficient Multi-Agent Learning Systems**

**Rating:** 7
**Confidence:** 4

**Review:**

* The paper studies multi agent reinforcement learning algorithms (MARL) to characterize the bottlenecks in both both cooperative and actor critic setting
* The authors find that as the number of agents scale up, the update all trainer step becomes the significant bottleneck taking upto 87% of the execution time for 48 agents.
* Among the various steps in the update trainers the mini-batch sampling stage takes the most time, followed by the target Q calculation step.
* The authors propose a new locality aware mini-batch sampling algorithm which reduces the time consumed in the mini-batch sampling step by 27.39% for 12 agents, which results in about 10.2% percent reduction in total training time.

**Review (Strengths/Weaknesses):**

* The characterization of the various MARL algorithms on real hardware revealing actionable points to improve.
* The study of hardware metrics like dTLB misses and cache misses are insightful.
* Improved algorithm for locality aware mini batch sampling operation

Additional points to improve
*  It is not clear that the proposed algorithm is eliminating the slowdown due to random accesses from the critical path or just delegating into a previous steps. Some additional analysis  on end-to-end runtime will be insightful
* There are typos in the draft which need correction. Eg. on sec VI, "Moreover, fetching the transition data from far way memory location"

**Reviewer Expertise:**

Knowledgeable: I used to work in this area and/or I try to keep up with the literature but might not know the latest developments.